# OpenReview forum: "Explicit Representation Alignment via Subspace Elimination for Robust LLM Unlearning"
_ICLR.cc/2026/Conference — Submitted to ICLR 2026_

### Official Review · Reviewer_Z9rN · 2025-10-31

**Soundness:** 3
**Presentation:** 3
**Contribution:** 2
**Rating:** 4
**Confidence:** 3

**Summary:**

This paper proposes ERASER (Explicit Representation Alignment via Subspace Elimination for Robust LLM Unlearning), a principled framework for stable and effective unlearning in large language models. ERASER identifies harmful directions in the representation space and nullifies them via a Subspace-based Target Vector, eliminating the need for heuristic tuning. It further introduces a Disentangle-Head, a ranking-based auxiliary module that separates forget and retain representations, improving convergence and stability.

**Strengths:**

1.  Compared with RMU’s random target vector, the Subspace-based Target Vector offers a clear advantage: It replaces heuristic randomness with a data-driven forgetting subspace, enabling more stable and interpretable unlearning across diverse datasets and models.

2.  Experiments on multiple LLMs show state-of-the-art forgetting performance with minimal loss in general ability and strong robustness against jailbreak, membership inference, and relearning attacks.

**Weaknesses:**

1.  A limitation of the Disentangle-Head is that it relies on both forget and retain samples, which may constrain its applicability when clear domain separation or labeled retain data is unavailable.

2.  The proposed method heavily depends on the RMU framework and is difficult to generalize to other unlearning paradigms such as gradient ascent or negative preference optimization.

3.  The evaluation is limited to the WMDP benchmark, lacking validation on other unlearning datasets such as TOFU, MUSE, and RWKU, which raises concerns about the generality of the results.

**Questions:**

Please see the above Weaknesses.

---

> ### Author Response · Authors · 2025-11-20
>
> We sincerely thank the reviewer for taking the time to provide a careful assessment of our work. Your insights on what is working well and what can be improved are deeply appreciated, and they will meaningfully guide the refinement of our paper.
>
> ---
>
> **W1 (domain separation or labeled retain data)**
>
> Following the reviewer's comment, **we investigated 54 recent papers (the detailed paper list is presented in the last comment) across NeurIPS, ICML, ACL, EMNLP, NAACL, CVPR, and ICCV.** We observed that 81% explicitly use a publicly available retain set. Such retain corpora (e.g., WikiText) require no labeling and can be used directly, which makes the assumption practical in real deployments. Importantly, ERASER does not require any domain-specific annotation for the retain set; only the forget domain must be defined and labeled. This setting corresponds to the typical real-world scenario where the harmful or sensitive domain is known, while general-purpose text (news, Wikipedia, books) is freely available and can serve as the retain set without additional supervision.
>
> ---
>
> **W2 (generalize to other unlearning paradigms)**
>
> Although our experiments instantiate ERASER on top of the RMU objective, the core mechanism, constructing a shared forgetting subspace and applying projection nullification at the representation level, is not specific to RMU. This projection step precedes the loss computation and is therefore compatible with other unlearning paradigms such as gradient ascent or negative preference optimization.
>
> At the same time, we acknowledge that demonstrating such integrations experimentally is beyond the scope of this work. **Our focus was to address a major open issue in RMU: the lack of a principled method for determining target vectors. We appreciate that Reviewer rEJU independently recognized this as a substantial contribution.** Reviewer rEJU noted that “Some might view this work as incremental since it builds on RMU and adjusts target vectors. However, the reviewer believes the contributions are substantial because there has been no principled method to determine target vectors, which is a key limitation of RMU. Overcoming this limitation is a significant advancement.”
>
> These positive observations indicate that ERASER’s main contribution, principled target-vector construction and representation-level disentanglement, is orthogonal to the specific choice of unlearning objective. While we did not experimentally combine ERASER with GA or NPO due to scope constraints, we agree that this is a valuable direction for future work and appreciate the reviewer’s suggestion.
>
> ---
>
> **W3 (beyond WMDP benchmark)**
>
> We appreciate the reviewer’s concern regarding generality. However, several widely referenced unlearning benchmarks are not directly aligned with the type of concept/knowledge-level harmful knowledge targeted by ERASER. For example, TOFU consists of autobiographies of 200 fictitious authors, and RWKU focuses on unlearning real-world individuals. Both represent entity-level unlearning, where removing an entity name or its direct associations does not require altering the underlying conceptual knowledge. As a result, these benchmarks are not well-suited for evaluating a subspace-based method aimed at forgetting deeper semantic content.
>
> Beyond WMDP, we additionally evaluate ERASER on the Fictional Knowledge Dataset, which explicitly probes whether forgetting extends beyond surface-form memorization. As shown in Table 3, ERASER exhibits strong forgetting across *memorization*, *semantic generalization*, and *compositional generalization*, demonstrating modification of deeper internal representations rather than simple output suppression.
>
> We also conducted experiments on the MUSE (News), as you suggested, which provides a more diverse evaluation through four complementary metrics: (1) verbatim memorization, (2) privacy leakage, (3) semantic knowledge memorization on the forget set, and (4) retained knowledge. Across all metrics, ERASER delivers a substantially better unlearning–utility trade-off than RMU. Notably, ERASER yields dramatically lower privacy leakage, competitive or stronger reductions in both verbatim and semantic memorization, and significantly higher retained knowledge, indicating more stable model utility.
>
> Taken together, evaluations on WMDP, Fictional Knowledge, and MUSE collectively show that ERASER generalizes beyond a single benchmark and consistently achieves strong forgetting while preserving useful capabilities. We will highlight these broader results in the updated version.
>
> | Method | verbmem_f ↓ | privleak ↓ | knowmem_f ↓ | knowmem_r ↑ |
> |--------|-------------|------------|-------------|-------------|
> | Base   | 18.32       | 22.08      | 27.24       | 35.32       |
> | ERASER | 22.62       | -98.95     | 32.62       | 31.75       |
> | RMU    | 22.81       | -80.80     | 33.17       | 28.82       |

---

> ### Author Response · Authors · 2025-11-20
>
> **Related to response [W1]**
>
> We compiled the relevant references for the prior work we discussed. After reviewing unlearning papers published in major conferences over the past two years, we observed that 44 out of 54 papers, representing about 81 percent, incorporated a retain set in their methodology. For ease of presentation, we provide the list of these 44 papers without author names.
>
>
> ---
>
> 1. The WMDP Benchmark: Measuring and Reducing Malicious Use with Unlearning (ICML 2024)
>
> 2. Soft Prompting for Unlearning in Large Language Models (NAACL 2025)
>
> 3. Large Language Model Unlearning (NeurIPS 2024)
>
> 4. LLM Unlearning via Embedding-Corrupted Prompts (NeurIPS 2024)
>
> 5. Reversing the Forget-Retain Objectives: Unlearning via Logit Difference (ULD) (NeurIPS 2024)
>
> 6. Machine Unlearning of Pre-trained Large Language Models (ACL 2024)
>
> 7. Unlearn What You Want to Forget: Efficient Unlearning for LLMs (EMNLP 2023)
>
> 8. Towards LLM Unlearning Resilient to Relearning Attacks (ICML 2025)
>
> 9. Unified Gradient-Based Machine Unlearning with Remain Geometry Enhancement (NeurIPS 2024)
>
> 10. Defensive Unlearning with Adversarial Training for Robust Concept Erasure in Diffusion Models (NeurIPS 2024)
>
> 11. Data Attribution for Text-to-Image Models by Unlearning Synthesized Images (NeurIPS 2024)
>
> 12. What makes unlearning hard and what to do about it (NeurIPS 2024)
>
> 13. Unveiling and Mitigating Backdoor Vulnerabilities based on Unlearning Weight Changes and Backdoor Activeness (NeurIPS 2024)
>
> 14. Certified Machine Unlearning via Noisy Stochastic Gradient Descent (NeurIPS 2024)
>
> 15. Boosting Alignment for Post-Unlearning Text-to-Image Generative Models(NeurIPS 2024)
>
> 16. Towards Unbounded Machine Unlearning (NeurIPS 2023)
>
> 17. Fast Model Debias with Machine Unlearning (NeurIPS 2023)
>
> 18. Shared Adversarial Unlearning: Backdoor Mitigation by Unlearning Shared Adversarial Examples (NeurIPS 2023)
>
> 19. Certified Minimax Unlearning with Generalization Rates and Deletion Capacity (NeurIPS 2023)
>
> 20. Efficient Source-free Unlearning via Energy-Guided Data Synthesis and Discrimination-Aware Multitask Optimization (ICML 2025)
>
> 21. Targeted Unlearning with Single Layer Unlearning Gradient (ICML 2025)
>
> 22. SAeUron: Interpretable Concept Unlearning in Diffusion Models with Sparse Autoencoders (ICML 2025)
>
> 23. PDUDT: Provable Decentralized Unlearning under Dynamic Topologies (ICML 2025)
>
> 24. A Certified Unlearning Approach without Access to Source Data (ICML 2025)
>
> 25. Leveraging Per-Instance Privacy for Machine Unlearning (ICML 2025)
>
> 26. When to Forget? Complexity Trade-offs in Machine Unlearning (ICML 2025)
>
> 27. Certified Unlearning for Neural Networks (ICML 2025)
>
> 28. Not All Wrong is Bad: Using Adversarial Examples for Unlearning (ICML 2025)
>
> 29. Knowledge Swapping via Learning and Unlearning (ICML 2025)
>
> 30. Towards Certified Unlearning for Deep Neural Networks (ICML 2024)
>
> 31. Verification of Machine Unlearning is Fragile (ICML 2024)
>
> 32. Decoupled Distillation to Erase: A General Unlearning Method for Any Class-centric Tasks (CVPR 2025)
>
> 33. LoTUS: Large-Scale Machine Unlearning with a Taste of Uncertainty (CVPR 2025)
>
> 34. NoT: Federated Unlearning via Weight Negation (CVPR 2025)
>
> 35. Towards Source-Free Machine Unlearning (CVPR 2025)
>
> 36. Unlearning through Knowledge Overwriting: Reversible Federated Unlearning via Selective Sparse Adapter (CVPR 2025)
>
> 37. Generative Unlearning for Any Identity (CVPR 2024)
>
> 38. MUNBa: Machine Unlearning via Nash Bargaining (ICCV 2025)
>
> 39. Unlearning the Noisy Correspondence Makes CLIP More Robust (ICCV 2025)
>
> 40. Robust Machine Unlearning for Quantized Neural Networks via Adaptive Gradient Reweighting with Similar Labels (ICCV 2025)
>
> 41. Meta-Unlearning on Diffusion Models: Preventing Relearning Unlearned Concepts (ICCV 2025)
>
> 42. Forgetting Through Transforming: Enabling Federated Unlearning via Class-Aware Representation Transformation (ICCV 2025)
>
> 43. FG-OrIU: Towards Better Forgetting via Feature-Gradient Orthogonality for Incremental Unlearning (ICCV 2025)
>
> 44. Learning to Unlearn while Retaining: Combating Gradient Conflicts in Machine Unlearning (ICCV 2025)

---

> > ### Comment · Reviewer_Z9rN · 2025-11-26
> >
> > Thank you for your response. Regarding the combination of ERASER with GA or NPO, are there any feasible approaches you would recommend? In addition, I do not see a substantial difference between the Fictional Knowledge Dataset and TOFU or RWKU. It similarly injects passages that contain descriptions of fictional entities.

---

> ### Author Response · Authors · 2025-11-26
>
> Thank you for your careful follow-up question and for engaging so thoroughly with our previous response. We sincerely appreciate the level of attention and consideration you have given to our work.
>
> ---
>
>
> **Regarding the combination of ERASER with GA or NPO, are there any feasible approaches you would recommend?**
>
> We agree that, as you pointed out, GA and NPO [1] are defined at the token–level prediction loss, whereas ERASER operates at the representation level on hidden states $h_{\theta^{(\ell)}}^{(\ell)}(x)$. This distinction actually makes the combination conceptually straightforward: ERASER can be added as an auxiliary representation-level regularizer on top of GA or NPO, without changing their input–output structure.
>
> *1. ERASER as a representation-level regularizer*
>
>
> In our formulation, for a forget sample $x_F$, we extract the hidden representation
>
> $$
> r_F = h_{\theta_{\mathrm{unl}}}^{(\ell)}(x_F),
> $$
>
> compute the shared forgetting subspace $V_{\mathrm{shared}}$ as in Eqs. (4)–(7), and define the nullified target
>
> $$
> v_{\mathrm{target}}(x_F) = r_F - (r_F V_{\mathrm{shared}}^\top) V_{\mathrm{shared}}.
> $$
>
> This induces the representation-level forgetting loss (Eq. (9)):
>
> $$ L_{forget} = E_{x_F \sim D_{forget}} \| (r_F V_{shared}^\top) V_{shared} \|_2^2 $$
>
> Crucially, this loss is computed on hidden states, not on the token input. The standard language-modeling or unlearning loss (GA or NPO) is still defined on $(x_F, y_F)$ in the usual way. Thus, we can add an extra constraint on how $r_F$ is allowed to evolve.
>
> *2. Combining ERASER with Gradient Ascent (GA)*
>
>
> The GA objective on the forget set is
>
> $$ L_{GA}(\theta) = E_{(x_F, y_F) \sim D_{forget}} [ \log \pi_\theta(y_F \mid x_F) ] $$
>
> To combine GA with ERASER, one can define a joint objective of the form
>
> $$
> L_{\mathrm{total}}^{\mathrm{GA+ERASER}}(\theta) = -L_{\mathrm{GA}}(\theta) + \lambda L_{\mathrm{forget}}(\theta; V_{\mathrm{shared}}) + \alpha L_{\mathrm{retain}}(\theta) + \beta L_{\mathrm{Disentangle-Head}}(\theta),
> $$
>
> where
>
> * $L_{\mathrm{forget}}$ uses $r_F$ and $v_{\mathrm{target}}(x_F)$ as defined above,
> * $L_{retain} = E_{x_R} \| h_{\theta_{unl}}^{(\ell)}(x_R) - h_{\theta_{frozen}}^{(\ell)}(x_R) \|_2^2$ (Eq. (2)),
> * $L_{\mathrm{Disentangle-Head}}$ is the Disentangle-Head loss in Eq. (10),
> * $\lambda, \alpha, \beta$ are weighting coefficients.
>
> In this view, GA still plays its usual role: it pushes up the prediction loss on the forget set. Also, our proposed objective function shapes the geometry of the hidden representations so that forget samples are pushed away from the shared forgetting subspace and separated from retain samples. Thus, the two components act at different levels (output vs. representation) and are mathematically compatible.
>
> *3. Combining ERASER with NPO*
>
>
> The NPO loss on the forget set is
>
> $$ L_{NPO,\beta}(\theta) = \frac{2}{\beta} E_{(x_F, y_F)} [ \log ( 1 + ( \frac{\pi_\theta(y_F \mid x_F)}{\pi_{ref}(y_F \mid x_F)} )^\beta ) ] $$
>
>
> Analogously, ERASER can be combined by using the same representation-level regularizers:
>
> $$
> L_{\mathrm{total}}^{\mathrm{NPO+ERASER}}(\theta) = L_{\mathrm{NPO},\beta}(\theta) + \lambda L_{\mathrm{forget}}(\theta; V_{\mathrm{shared}}) + \alpha L_{\mathrm{retain}}(\theta) + \beta L_{\mathrm{Disentangle-Head}}(\theta).
> $$
>
> Again, NPO operates at the level of tokens, while ERASER constrains the latent space via projection-based modifications to the hidden states.
>
> From this perspective, ERASER is agnostic to the specific unlearning objective: it provides a representation-level mechanism (shared forgetting subspace + disentanglement) that can, in principle, be combined with GA or NPO formulations by simply adding $L_{\mathrm{forget}}$, $L_{\mathrm{retain}}$, and $L_{\mathrm{Disentangle-Head}}$ as auxiliary terms on top of the chosen unlearning loss.
>
> However, we note that the practical effectiveness of such combinations remains uncertain, because methods like GA are known to have fundamental optimization limitations—prior work [2] has shown that pure its objectives often converge poorly or destabilize retention, even without additional auxiliary terms. Thus, while the combination is conceptually feasible, its empirical behavior would require careful study beyond the scope of this work.
>
> [1] Zhang et al., 2024, Negative Preference Optimization: From Catastrophic Collapse to Effective Unlearning
>
> [2] Kim et al., 2025, Unlearning-Aware Minimization

---

> > ### Author Response · Authors · 2025-11-26
> >
> > **In addition, I do not see a substantial difference between the Fictional Knowledge Dataset and TOFU or RWKU. It similarly injects passages that contain descriptions of fictional entities.**
> >
> > While the Fictional Knowledge Dataset also contains descriptions of fictional entities, its evaluation protocol and knowledge structure differ fundamentally from TOFU and RWKU. TOFU and RWKU operate strictly at the entity level: the forget set is defined as a set of individuals (either fictitious authors or real-world people), and the evaluation probes focus on suppressing information tied to those specific entities. In these settings, the model can succeed largely by reducing the association between a name and its attributes, without altering the broader conceptual knowledge that the model has learned.
> >
> > In contrast, the Fictional Knowledge Dataset is explicitly designed to test multi-level knowledge generalization, not entity suppression. The dataset evaluates forgetting along three increasingly abstract dimensions: (1) memorization, (2) semantic generalization, and (3) compositional generalization. Crucially, the semantic and compositional probes require understanding and manipulating relations between concepts, rather than recalling or suppressing attributes of a single entity. Forgetting must therefore occur at the concept/knowledge level, not merely through name-based blocking or entity removal.
> >
> > This structural difference is why the Fictional Knowledge Dataset is better aligned with ERASER’s goal of forgetting harmful representational subspaces shared across domains. Concept-level forgetting requires modifying internal representations, exactly what ERASER is designed to do, whereas TOFU and RWKU can be addressed only through comparatively entity-level suppression.
> >
> > We hope this clarifies why the Fictional Knowledge Dataset provides a more appropriate testbed for evaluating concept-level unlearning. We also note that ERASER performs well on the MUSE benchmark, further supporting its generality.

---

### Official Review · Reviewer_rEJU · 2025-10-31

**Soundness:** 4
**Presentation:** 4
**Contribution:** 3
**Rating:** 8
**Confidence:** 3

**Summary:**

This paper presents ERASER, a representation-based LLM unlearning framework that identifies and neutralizes harmful directions in forget representations, building on RMU (Li et al., 2024). Two main ideas are introduced: first, by creating a shared forgetting subspace using domain-specific principal components and projecting forget-sample activations away from this subspace, ERASER stabilizes unlearning without relying on heuristics. Second, it introduces a Disentangle-Head auxiliary ranking loss that speeds up the separation between forget and retain domains. Extensive experiments demonstrate state-of-the-art forgetting on benchmarks, more thorough semantic erasure of fictional knowledge, and improved robustness against jailbreak, membership inference, and relearning attacks.

**Strengths:**

- Two novel ideas (shared forgetting subspace and Disentangle-Head loss) are introduced to enhance representation-based LLM unlearning. These ideas are well-motivated both empirically and logically.
- Extensive experiments, including comparisons with existing methods using four types of LLMs on standard datasets in LLM unlearning, along with ablation studies, demonstrate the effectiveness of ERASER across various benchmarks and attack scenarios. The ablation studies convincingly show the contribution of two components to the overall performance.

**Weaknesses:**

- One natural variation of the proposed method could be explored. Basically, the paper's idea is to push representations away from certain directions (forgetting subspace), which is estimated from forget samples from all domains. Another natural idea is to leverage a domain-wise forgetting subspace, i.e., estimating a forgetting subspace for each domain separately (use $\mathbf{Z}_j$ directly?) and pushing representations away from the corresponding domain-specific forgetting subspace. It would be interesting to see how this variation performs compared to the proposed method. If the reviewer understands correctly, the number of forgotten domains in the experiments is only two (because it has WMDP-Bio and WMDP-Cyber), so this variation would be feasible (and theoretically, it should work better?).
- While RMU collapses all the forget representations into a single (random) target vector, ERASER projects representations away from a forgetting subspace, which means that forget representations will be dispersed in the representation space. This creates a significant difference in the unlearning objective compared to RMU, and it would strengthen the paper if the authors could provide more analysis of this difference, both theoretically and empirically, to better understand the effectiveness (and possible downside) of the proposed approach.
- Some might view this work as incremental since it builds on RMU and adjusts target vectors. However, the reviewer believes the contributions are substantial because there has been no principled method to determine target vectors, which is a key limitation of RMU. Overcoming this limitation is a significant advancement.

**Questions:**

N/A

---

> ### Author Response · Authors · 2025-11-20
>
> We sincerely thank the reviewer for the exceptionally generous and encouraging assessment of our work. Your positive feedback and constructive suggestions are greatly appreciated and have helped us further strengthen the manuscript.
>
>
> ---
>
> **W1 (leverage a domain-wise forgetting subspace)**
>
> Thank you for the insightful suggestion. We implemented the domain-wise variation you proposed, constructing a separate forgetting subspace for each domain and pushing each sample away from its corresponding domain-specific subspace. The results are:
>
> | Method                               | MMLU   | WMDP-Bio | WMDP-Cyber |
> |--------------------------------------|--------|----------|------------|
> | Domain-wise subspace (proposed)      | 0.5684 | 0.2887   | 0.2597     |
> | Ours (shared forgetting subspace)  | 0.5662 | 0.2765   | 0.2481     |
>
> The performance is highly similar across all metrics, and in fact, the shared subspace performs slightly better on the forget domains. This empirical finding suggests that the domain-specific subspaces often overlap substantially in practice, especially for WMDP-Bio and WMDP-Cyber, so constructing a unified shared forgetting subspace provides a more robust and stable representation of harmful directions. In this sense, your suggestion helped reinforce the validity of our design choice. We will include this comparison in the revised version.
>
>
> ---
>
> **W2 (theoretical analysis)**
>
>
> We thank the reviewer for the insightful suggestion highlighting the geometric difference between RMU and ERASER. Motivated by this comment, we established a new theoretical comparison between the specific RMU loss and our subspace-based objective in the Theorem. 1–2 (please see below the comment).
>
> Although these theorems focus on special cases of RMU relative to our subspace method, they yield the following insights:
>
> - Our method can be viewed as a generalized version of RMU with a zero-mean random unit target vector, and it *reduces* to RMU when the forgetting subspace spans the entire space $\\mathbb{R}^d$.
> - Moreover, the expected RMU objective is globally isotropic with curvature $2I_d$, whereas the ERASER objective has Hessian $2\\mathbf{P}$. Concretely, RMU enforces a *global* collapse of representations toward a single attractor in all directions, while ERASER enforces collapse only within the forgetting subspace and is intentionally indifferent to the geometry in the orthogonal complement. This can be further analyzed with Hessian functions, similar to sharpness-aware minimization [1, 2]
>
> We again thank the reviewer for providing this insight, and we will definitely incorporate this discussion into the main text.
>
>
> ---
>
> **W3 (Overcoming this limitation is a significant advancement)**
>
> We fully agree with this assessment. In our own practical experience applying RMU, we observed that its behavior is indeed highly sensitive to the choice of target vector, often leading to instability and inconsistent unlearning quality in real deployments. We therefore appreciate the reviewer’s recognition that introducing a principled, data-driven mechanism for determining target vectors directly addresses one of RMU’s central limitations. This was precisely the motivation behind ERASER, and we are grateful that the reviewer identified this contribution clearly. In the camera-ready version, we will further emphasize this point to clarify why principled target construction is essential and how our approach resolves the core instability observed in RMU.
>
> ---
>
> [1] Foret, Pierre, et al. "Sharpness-aware minimization for efficiently improving generalization." ICLR 2021 Spotlight (2020).
>
> [2] Kwon, J., Kim, J., Park, H., & Choi, I. K. (2021, July). Asam: Adaptive sharpness-aware minimization for scale-invariant learning of deep neural networks. In *International conference on machine learning* (pp. 5905-5914). PMLR.

---

> ### Author Response · Authors · 2025-11-20
>
> **[Theorem 1]** Our subspace method can be viewed as a general formulation of RMU with the predefined vector to $u = 0$
>
> ---
> **Theorem 1**
> (Deterministic target $u = 0$).
>
> Let $d \\in \\mathbb{N}$ and let $\\mathbf{V}_{\\text{shared}} \\in \\mathbb{R}^{k \\times d}$ have orthonormal rows, so that
>
> $$
> \\mathbf{V} _ {\\mathrm{shared}} \\mathbf{V} _ {\\mathrm{shared}}^{\\top} = I_k.
> $$
>
> Define the orthogonal projector
>
> $$
> \\mathbf{P} := \\mathbf{V} _ {\\mathrm{shared}}^{\\top}\\mathbf{V} _ {\\mathrm{shared}} \\in \\mathbb{R}^{d \\times d},
> $$
>
> and consider, for any $\\mathbf{r} \\in \\mathbb{R}^{1 \\times d}$, the forgetting objective
>
> $$
> L_{\\text{forget}}(\\mathbf{r}) := \\|\\mathbf{r} \\mathbf{P}\\|_2^2 = \\mathbf{r} \\mathbf{P} \\mathbf{r}^{\\top}.
> $$
>
> For a deterministic target $\\mathbf{u} = 0$, define
>
> $$
> L_0(\\mathbf{r}) := \\|\\mathbf{r}\\|_2^2.
> $$
>
> Then $L_{\\text{forget}}(\\mathbf{r}) = L_0(\\mathbf{r})$ for all $\\mathbf{r}$ if and only if
>
> $$
> \\mathbf{P} = I_d,
> $$
>
> that is, the shared forgetting subspace spans the entire space $\\mathbb{R}^d$ (equivalently, $k=d$ and $\\mathbf{V}_{\\text{shared}}$ is orthogonal).

---

> ### Author Response · Authors · 2025-11-20
>
> **[Theorem 2]** Our subspace method can be viewed as a general formulation of RMU with the random unit vector (not-fixed)
>
> ---
>
> **Theorem 2** (Zero-mean random target).
>
> Let $d \\in \\mathbb{N}$ and $\\mathbf{V} _ {\\mathrm{shared}} \\in \\mathbb{R}^{k \\times d}$ have orthonormal rows, with associated projector
>
> $$
> \\mathbf{P} := \\mathbf{V} _ {\\mathrm{shared}}^{\\top} \\mathbf{V} _ {\\mathrm{shared}}.
> $$
>
> For $\\mathbf{r} \\in \\mathbb{R}^{1 \\times d}$, define
>
> $$
> L _ {\\mathrm{forget}}(\\mathbf{r}) := \\|\\mathbf{r}\\mathbf{P}\\|_2^2.
> $$
>
> Let $\\mathbf{u}$ be a random vector in $\\mathbb{R}^{1 \\times d}$ with finite second moment and zero-mean
>
> $$
> \\boldsymbol{\\mu} := \\mathbb{E}[\\mathbf{u}] = 0.
> $$
>
> Define the expected squared-distance objective
>
> $$
> \\tilde{L}(\\mathbf{r}) := \\mathbb{E} _ {\\mathbf{u}} [\\|\\mathbf{r} - \\mathbf{u}\\|_2^2].
> $$
>
> The function $\\tilde{L}$ can be rewritten as:
>
> $$
> \\tilde{L}(\\mathbf{r}) = \\|\\mathbf{r}\\|_2^2 - 2\\boldsymbol{\\mu}\\mathbf{r}^{\\top} + C,
> $$
>
> where $C := \\mathbb{E}[\\|\\mathbf{u}\\|_2^2]$. In particular,
>
> $$
> \\nabla _ {\\mathbf{r}}\\tilde{L}(\\mathbf{r}) = 2(\\mathbf{r} - \\boldsymbol{\\mu}), \\qquad \\nabla _ {\\mathbf{r}}^2\\tilde{L}(\\mathbf{r}) = 2I_d.
> $$
>
> When $\\mathbf{P} = I_d$ and $\\boldsymbol{\\mu} = 0$, there exists a constant $C' \\in \\mathbb{R}$ such that
>
> $$
> L _ {\\mathrm{forget}}(\\mathbf{r}) = \\tilde{L}(\\mathbf{r}) + C' \\qquad \\text{for all } \\mathbf{r} \\in \\mathbb{R}^{1 \\times d}.
> $$
>
> that is, the shared forgetting subspace spans the entire space $\\mathbb{R}^d$ (equivalently, $k=d$ and $\\mathbf{V} _ {\\mathrm{shared}}$ is orthogonal).
>
>
>
> **Proof.** Assuming $L _ {\\mathrm{forget}}(\\mathbf{r}) = \\tilde{L}(\\mathbf{r}) + C'$ for all $\\mathbf{r}$,
>
> $$
> \\mathbf{r}\\mathbf{P}\\mathbf{r}^{\\top} = \\|\\mathbf{r}\\|_2^2 - 2\\boldsymbol{\\mu}\\mathbf{r}^{\\top} + (C + C').
> $$
>
> Taking gradients with respect to $\\mathbf{r}$ gives
>
> $$
> 2\\mathbf{r}\\mathbf{P} = 2(\\mathbf{r} - \\boldsymbol{\\mu}) \\qquad \\forall \\mathbf{r}.
> $$
>
> Differentiating again,
>
> $$
> 2\\mathbf{P} = \\nabla _ {\\mathbf{r}}^2 L _ {\\mathrm{forget}}(\\mathbf{r}) = \\nabla _ {\\mathbf{r}}^2 \\tilde{L}(\\mathbf{r}) = 2I_d,
> $$
>
> hence $\\mathbf{P} = I_d$. Substituting into the gradient identity,
>
> $$
> \\mathbf{r} = \\mathbf{r} - \\boldsymbol{\\mu} \\qquad \\forall \\mathbf{r},
> $$
>
> implies $\\boldsymbol{\\mu} = 0$. Conversely, if $\\mathbf{P} = I_d$ and $\\boldsymbol{\\mu} = 0$, then
>
> $$
> L _ {\\mathrm{forget}}(\\mathbf{r}) = \\|\\mathbf{r}\\|_2^2, \\qquad \\tilde{L}(\\mathbf{r}) = \\|\\mathbf{r}\\|_2^2 + C,
> $$
>
> so the two objectives differ only by an additive constant. This completes the proof.

---

### Official Review · Reviewer_tez5 · 2025-10-31

**Soundness:** 3
**Presentation:** 3
**Contribution:** 3
**Rating:** 6
**Confidence:** 4

**Summary:**

This paper proposes ERASER, a representation-guided unlearning framework for large language models (LLMs), designed to effectively and stably remove harmful knowledge from models while preserving their general language capabilities. The core innovations of ERASER include:1.A Subspace-based Target Vector construction, which identifies and nullifies harmful directions in the representation space.2.An auxiliary module, the Disentangle-Head, which promotes the separation of representations between the forget and retain domains via a ranking loss.
The authors validate ERASER's effectiveness on multiple base models (Zephyr, Qwen, Llama, Yi) and conduct a comprehensive evaluation covering unlearning efficacy, semantic-level forgetting, and robustness against adversarial attacks. The results demonstrate that ERASER achieves the best balance between targeted forgetting and knowledge retention.

**Strengths:**

1.The proposed data-driven approach for constructing the target vector via subspaces avoids the limitations of existing methods that rely on heuristic targets, significantly enhancing the stability and generalizability of unlearning.
2. Beyond traditional unlearning assessments, the paper provides the first systematic evaluation of model robustness at semantic/compositional levels and against various adversarial threats (e.g., Jailbreak, Membership Inference, Relearning attacks), validating the method's practical utility.
3.The Disentangle-Head is demonstrated to be a general-purpose plugin that can improve the performance of other unlearning methods (e.g., RMU), enhancing the method's versatility and practical value.

**Weaknesses:**

1.While the Fictional Knowledge dataset is useful for evaluating deep semantic forgetting, there might be a gap between it and the complex, implicit harmful knowledge in the real world. The method's generalization needs further verification in more realistic and diverse harmful knowledge scenarios.
2. Although the parameter efficiency advantage of ERASER is mentioned, the computational cost of the offline subspace construction phase is not analyzed in detail, especially regarding its scalability in large-scale, multi-domain forgetting scenarios.
3.Although the method performs excellently in experiments, there is a lack of theoretical or interpretability analysis regarding "to what extent forgetting is considered successful," making it difficult to assess the risk of over-forgetting or under-forgetting.

**Questions:**

1.The construction of ERASER's shared forgetting subspace relies on a predefined number of forget domains N and principal components k. In practical applications where forget samples come from unknown or dynamically changing domains, how can the method adaptively determine these hyperparameters?
2.In the adversarial evaluation, ERASER shows robustness against Relearning Attacks. However, if an adversary uses a larger-scale or more representative subset of the forget data for retraining, would the model's unlearning effectiveness degrade significantly?

---

> ### Author Response · Authors · 2025-11-20
>
> We sincerely thank the reviewer for the thoughtful evaluation and constructive comments. Your feedback has helped us improve the clarity and overall quality of the paper.
>
> ---
>
> **W1 (diverse harmful knowledge scenarios)**
> We agree that the Fictional Knowledge dataset, while useful for probing deeper forms of semantic forgetting, does not fully capture the complexity of implicit harmful knowledge in real-world settings. At present, to the best of our knowledge, the community lacks a benchmark that comprehensively reflects such broad, context-dependent harmful domains, making this an open challenge for future unlearning research.
>
> We also note that recent efforts, such as RWKU, have attempted to incorporate real-world realism by using actual individuals. However, it still only reflects entity-level unlearning, removing information tied to specific names, rather than the richer concept-level harmful knowledge that ERASER aims to suppress. Thus, it also does not fully address the reviewer’s concern.
>
> Moreover, to broaden the evaluation beyond fictional and WMDP-style scientific knowledge, **we additionally evaluated ERASER on the MUSE (News) benchmark, which captures more realistic distributions through news-style articles and reports.** In this setting, ERASER demonstrated a substantially stronger unlearning utility than RMU, showing lower privacy leakage and competitive reductions in memorization while retaining more useful knowledge.
>
> | Method | verbmem_f ↓ | privleak ↓ | knowmem_f ↓ | knowmem_r ↑ |
> |--------|-------------|------------|-------------|-------------|
> | Base   | 18.32       | 22.08      | 27.24       | 35.32       |
> | ERASER | 22.62       | -98.95     | 32.62       | 31.75       |
> | RMU    | 22.81       | -80.80     | 33.17       | 28.82       |
>
> Nevertheless, developing richer and more diverse benchmarks for concept-level harmful knowledge remains a meaningful direction for future work, and we plan to pursue this as part of our broader research agenda.
>
> ---
>
> **W2 (computational cost)**
> Thank you for the observation. The offline subspace construction relies on power iteration, and its cost is modest in practice. As shown in Figure 2(b), ERASER is stable across a wide range of component counts, and a small number of directions (e.g., k) is sufficient, keeping computation lightweight. The cost scales linearly with the number of forget domains N, which is expected in any domain-specific unlearning setting. Since each domain must be processed at least once, this scaling is inherent to the problem rather than a limitation of our method. We will clarify this behavior in the camera-ready version.
>
> ---
>
> **W3 (to what extent forgetting is considered successful)**
> We agree that defining “how much forgetting is sufficient” remains an open problem in approximate unlearning. As noted in prior work on weak unlearning [1, 2, 3], the field does not yet have a formally accepted criterion or theoretical guarantee for quantifying successful forgetting, and most existing approaches, including RMU and SalUn, are evaluated empirically rather than through certified bounds. ERASER follows this widely adopted approximate-unlearning paradigm.
>
> To provide as much interpretability as is currently feasible, we report multiple complementary indicators: Forget, Utility, semantic-level forgetting, and robustness to relearning and jailbreak attacks, which together help assess both under- and over-forgetting in practice. We hope that a more rigorous metric for evaluating forgetting will be developed, and we will also make efforts toward contributing to that direction.
>
> ---
>
>
> [1] Fan et al., 2023, Salun: Empowering machine unlearning via gradient-based weight saliency in both image classification and generation.
>
> [2] Jia et al., 2023, Model sparsity can simplify machine unlearning
>
> [3] Madry et al., 2017, Towards deep learning models resistant to adversarial attacks

---

> ### Author Response · Authors · 2025-11-20
>
> **Q1 (hyperparameters)**
> We thank the reviewer for raising this important question. We would like to note that the relevant hyperparameter analyses are already included in the Appendix.
>
> - Number of forget domains (N):
>     - ERASER is evaluated with 2 forget domains and shows stable behavior.
>     - Appendix C.5 (Table 14): We additionally examine a 3-domain setting (e.g., Economics, Law, Physics), where forgetting remains effective.
> - Number of principal components (k):
>     - Figure 2(b): ERASER remains stable across a wide range of k values, indicating that extensive tuning is typically unnecessary.
>
> In practical deployments, the forget domains must first be defined by the application—unlearning cannot be performed meaningfully without specifying what information should be removed. ERASER does not assume a fixed number of domains and can operate with any N; it simply constructs the shared subspace from whichever domains the user provides.
>
> When adaptive adjustment of k is desired, the computational overhead is modest. As shown in Appendix B (Table 7), our method runs completely in about 4.5 minutes on a single 80GB GPU.
>
> ---
> **Q2 (relearning attacks)** We agree that, in principle, retraining on a very large portion of the original forget set could reintroduce knowledge, simply because this scenario becomes equivalent to an additional supervised fine-tuning stage. However, this setup falls outside the standard definition of a relearning attack used in the literature. Existing works [4, 5] assume realistic adversaries with limited access to the forget data, typically using only 20–60 samples—exactly the regime we evaluate. Under this established threat model, ERASER remains robust, as demonstrated in our experiments.
>
> Scaling the attack to tens of thousands of samples effectively transforms the scenario into full task-specific fine-tuning, not a targeted relearning attack. Thus, we believe that the robustness guarantees of ERASER should be interpreted within the commonly accepted adversarial setting, where our method performs consistently well.
>
> ---
>
> [4] Fan et al., 2025 Towards LLM Unlearning Resilient to Relearning Attacks: A Sharpness-Aware Minimization Perspective and Beyond
>
> [5] Wu et al., 2025 Beyond Sharp Minima: Robust LLM Unlearning via Feedback-Guided Multi-Point Optimization

---

### Official Review · Reviewer_oeex · 2025-11-01

**Soundness:** 2
**Presentation:** 3
**Contribution:** 2
**Rating:** 4
**Confidence:** 4

**Summary:**

The paper proposes ERASER, a representation-guided unlearning method for LLMs. It (i) computes a Shared Forgetting Subspace by aggregating domain-wise principal directions of forget sets and (ii) performs projection–nullification during training to remove those components from hidden states. An auxiliary Disentangle-Head adds a pairwise ranking loss to separate retain/forget representations and speed convergence. Empirically, ERASER improves ΔAcc over baselines across several base models and reports robustness to jailbreak, membership inference, and relearning attacks.

**Strengths:**

- **Principled target construction.** The subspace approach replaces heuristic/random target vectors, reducing sensitivity to hand-tuned coefficients.
- **Clear mechanics & reproducibility.** The three-phase pipeline and training objective are explicit, with algorithmic detail in the appendix.
- **Broad empirical scope.** Multi-model comparisons and robustness evaluations (GCG, embedding-space attacks, MIAs, relearning).

**Weaknesses:**

**W1. Baseline Reproducibility Issue (Adaptive RMU).** The reported Adaptive RMU[1]  performance diverges from the numbers in its original paper. The authors acknowledge variability but fail to reconcile the cause or show matched configurations. This undermines fairness in comparison and raises questions about the experimental setup consistency.

**W2. Missing optimization based Baselines.** The study omits comparisons to recent finetuning-style unlearning methods (e.g., Sim**NPO[2]**, **Gradient Ascent**). These are strong, widely-used baselines for LLM unlearning; excluding them weakens the empirical claim of superiority.

**W3. Limited Utility Evaluation.** The paper reports mainly **MMLU** results, lacking broader coverage of reasoning tasks (e.g., **GSM8K**, **MATH**, **BBH**) or multi-turn dialogue evaluations such as **MT-Bench**. Since unlearning often impacts reasoning ability, broader utility tests are necessary.

**W4. Insufficient Hyperparameter Sensitivity Analysis.** The paper reports fixed values (e.g., α = 1200) without explaining sensitivity. The wide gap between α for ERASER and for RMU (1–5) demands justification. Also, I did not find the definition of β.

**W5. Limited Exploration of Alternative Subspace Methods.** The method relies solely on PCA/SVD to define harmful directions. Variants like weighted PCA, robust PCA, or LEACE-style concept erasure are not tested, limiting generality.

**W6.  Ambiguous Relation to Existing Paradigms.** Conceptually, I see ERASER resembles “null-space optimization + DPO-style ranking”, which is actually common in knowledge editing [3]. The paper should clarify conceptual distinctions and empirical advantages over these established approaches. Especially explaining why do you need the auxilary loss.



[1] On Effects of Steering Latent Representation for Large Language Model Unlearning

[2]  Simplicity Prevails: Rethinking Negative Preference Optimization for LLM Unlearning

[3] AlphaEdit: Null-Space Constrained Knowledge Editing for Language Models

**Questions:**

**Q1. Unexplained Data Efficiency.** Why ERASER achieves similar performance using only 10 % of the forget set as with 100 %. This strong claim lacks analysis. It remains unclear whether the model genuinely generalizes from a small forget sample or simply overfits shared latent directions?

**Q2. Potential Over-forgetting due to Subspace Overlap.** If retain and forget domains overlap, does projection–nullification inadvertently erase useful features? Can you quantify retention degradation under controlled overlap?

**Q3. Attack Surface.** Could the shared subspace concentration make ERASER more predictable or vulnerable? Have you visualized the principal directions to verify domain diversity? Are jailbreak defenses mainly due to parameter updates or orthogonalization? Could adaptive attackers exploit complementary subspaces?

**Q4. Concept-level and Large-scale Forgetting.** Would ERASER remain effective on **book-scale (MUSE)** or **concept-level (RWKU)** unlearning tasks where harmful content spans broader contexts?

**Q5. Optimization Claim for joint forget/retain optimization.** You state that joint forget/retain optimization stability is an open problem `L106-L107`, yet you still jointly optimize. What empirical evidence (e.g., convergence variance, loss curvature) supports ERASER’s improved stability?

**Q6. The Ablation Study Discrepancies.** Why do Table 11 and 12 ablation results differ from the main experiments? Are the datasets or hyperparameters identical?

**Q7. Utility Drop Explanation In Appendix C3.** Why does full fine-tuning reduce general capabilities so drastically? Can you link this to your subspace framework (e.g., low-rank vs. full-rank drift)? It would be interesting if you have explanation about this phenomenon.

**Q8. Alternatives for the disentangle-head and subspace construction.**

- Would a simpler contrastive or orthogonal-loss baseline achieve similar disentanglement without the Disentangle-Head?
- If PCA were replaced by a weighted or robust variant, or if the subspace were built using mutual-information weighting instead of SVD, how would performance change?

---

> ### Author Response · Authors · 2025-11-20
>
> We sincerely thank the reviewer for the careful and constructive assessment of our work. Your detailed comments and suggestions are highly valuable and have helped us significantly improve the clarity and quality of the paper.
>
> ---
>
> **W1 (AdaptiveRMU)** We would like to note that we have made every effort to reproduce the results of AdaptiveRMU and did not use any unfair or modified configurations. **We strictly followed the settings proposed in the original paper** and its official GitHub repository, but the results could not be reproduced. As shown in Appendix D.1, we ran five different random seeds and observed substantial variance.
>
> ---
>
> **W2 (Optimization based Baselines)** Thank you for your suggestion. Following your suggestion, we provide a new table as follows:
>
> | Method | WMDP-Bio | WMDP-Cyber | MMLU |
> |--------|----------|------------|------|
> | Base   | 0.637    | 0.440      | 0.581 |
> | GA     | 0.458    | 0.202      | 0.545 |
> | NPO    | 0.425    | 0.283      | 0.400 |
> | SimNPO | 0.416    | 0.322      | 0.471 |
> | ERASER | 0.277    | 0.248      | 0.566 |
>
> As shown in the results, ERASER achieves more stable forgetting on both WMDP-Bio and WMDP-Cyber, while also preserving MMLU performance more reliably than the baselines.
>
>
> ---
>
> **W3 (Utility Evaluation)** Thank you for the suggestion. Following the reviewer’s recommendation, we additionally evaluated ERASER on GSM8K, MATH, BBH, and MT-Bench. Across GSM8K, MATH, and BBH, the performance of ERASER closely matches that of the base model, indicating that our method does not adversely affect core reasoning capabilities.
>
> | Method | GSM8K (5-shot) | BBH (3-shot) | MATH | MT-Bench |
> |--------|-----------------|--------------|-------|-----------|
> | Base   | 32.98     | 52.20    | 1.00   | 7.34      |
> | ERASER | 32.22     | 51.28  | 0.94  | 6.73      |
>
> For MT-Bench, we observe a modest drop of about 0.6 points. A closer breakdown shows that ERASER maintains strong single-turn performance, while the decrease arises primarily in multi-turn settings. This insight reveals that multi-turn dialogue may be more sensitive to unlearning, and we will definitely consider targeted mitigation of this effect as an important direction for future work.
>
> ---
>
> **W4 (Hyperparameter Sensitivity Analysis)** The definition of β is stated at line 239 as the hyperparameter of the Disentangle-Head loss, and we will add the explicit formula in the camera-ready version for clarity. The sensitivity of α is reported in Appendix Table 10. Across a wide range of α values, the variance in performance is small, indicating that ERASER is stable with respect to α. We followed the same α setting (1200) used in RMU for fair comparison, which explains why the numerical scale appears different despite serving an equivalent balancing role.
>
> ---
>
> **W5 (Alternative Subspace Methods)** Our contribution does not hinge on using PCA/SVD specifically, but on leveraging a dimension-reduction-based formulation to identify shared harmful directions. Among various alternatives, we selected power iteration due to its strong empirical performance and low computational cost, which is important for large-scale LLMs. We agree that other variants are compatible with our framework. To confirm that the performance of ERASER remains consistent regardless of the specific decomposition method, we conducted additional experiments as you stated. The results of these experiments can be found in Q8.
>
> ---
>
> **W6 (Ambiguous Relation to Existing Paradigms)** While ERASER includes a projection step, its objective is fundamentally different from null-space–based knowledge editing methods such as AlphaEdit, which operate in parameter space and manipulate explicit key–value associations. ERASER instead works entirely in the representation space and removes knowledge-level latent structure without relying on any query-specific or sentence-level editing mechanism. Likewise, although the Disentangle-Head uses a ranking form, it is not a DPO-style preference loss. DPO requires query, pos, and neg triplets tied to the same prompt, whereas ERASER is query-free and uses only forget and retain pairs drawn from different domains to enforce domain-level separation in latent space, an approach unique to unlearning. This auxiliary loss explicitly drives retain representations upward and forget representations downward, a mechanism that, to our knowledge, has not been used in prior unlearning work. **We would like to emphasize that Reviewer rEJU mentioned, "ERASER also provides the first principled alternative to RMU’s heuristic target vector, a contribution that is orthogonal to knowledge-editing or preference-optimization paradigms."**

---

> ### Author Response · Authors · 2025-11-20
>
> **Q1 (Unexplained Data Efficiency)**
>
> As shown in Appendix Table 13, randomly subsampling the forget set (e.g., using only 10%) yields performance comparable to using the full dataset. This is an inherent advantage of ERASER: once a sufficient number of examples is observed, the dominant harmful directions are stably identified, and additional samples offer diminishing returns.
>
> To verify that this effect is not caused by overfitting to any particular subset, we conducted an additional stability analysis. Specifically, we drew three independent subsets (A, B, C) of 100 forget samples each and computed their dominant principal directions using power iteration. **The pairwise cosine similarity between these directions is extremely high (|cos| ≈ 0.995), demonstrating that the harmful direction is highly stable across subsets, despite the small sample size.** This confirms that ERASER captures a genuine shared latent structure rather than overfitting to specific samples.
>
> | cosine sim |   A   |   B   |   C   |
> |------------|-------|-------|-------|
> | A          | 1.000     | 0.995 | 0.996 |
> | B          | 0.995 | 1.000     | 0.996 |
> | C          | 0.996 | 0.996 | 1.000     |
>
> These results show that different subsets drawn from the same forget distribution yield nearly identical principal directions. Thus, ERASER requires only a modest fraction of forget samples to recover the shared harmful direction, explaining why 10% achieves performance comparable to 100%.
>
> ---
>
> **Q2 (Potential Over-forgetting due to Subspace Overlap)**
>
> We agree that overlap between the retain and forget domains can potentially cause over-forgetting. To examine this, we conducted a controlled evaluation in Appendix C.5 (Table 14). Specifically, we used Economics, Law, and Physics as forget domains, and then measured retention on Econometrics, Jurisprudence, and Math, which are semantically aligned with the corresponding forget domains. As shown, ERASER effectively reduces performance on the forget domains, but retention is partially degraded when the semantic overlap is high.
>
> This behavior is consistent with prior work and reflects a known limitation of current unlearning methods. We believe handling fine-grained domain overlap is an important direction for future research in this area.
>
> ---
>
> **Q3 (Attack Surface)**
>
> Thank you for sharing this interesting idea. Current jailbreak attacks generally operate in a purely I/O–level, gradient-free manner and do not access internal representations or latent directions [1]. Therefore, a direct adaptive attack cannot be applied to our model, as we only modify the loss function, and the Disentangle-Head used during training is discarded afterward.
>
> Although our approach is an unlearning method rather than a defense mechanism against jailbreak attacks, we agree that it may offer meaningful insights regarding the robustness of LLMs. Since this direction is somewhat outside the main scope of our current work, we plan to explore it further in future research.
>
>
> ---
>
> **Q4 (Concept-level and Large-scale Forgetting)**
>
> We would like to clarify that RWKU does not evaluate concept-level forgetting. Its forget set consists of 200 real-world individuals from Wikipedia, and the probes mainly test whether the model can suppress information tied directly to those specific entities. This corresponds to entity-level unlearning, where forgetting is achieved by severing associations with an entity name rather than by removing the underlying conceptual knowledge. Because ERASER is designed to suppress broader, domain-level harmful knowledge, RWKU does not directly measure the form of semantic forgetting targeted by our method.
>
> In contrast, the Fictional Knowledge Dataset provides a structured hierarchy of memorization, semantic generalization, and compositional generalization, capturing progressively more abstract and concept-like forms of knowledge. As shown in Table 3, ERASER suppresses information consistently across all levels. From a knowledge-abstraction standpoint, this dataset more closely aligns with concept-level forgetting, and our results indicate that ERASER remains effective in this setting.
>
> **However, motivated by the reviews' comment, we conducted additional experiment on the MUSE (News) benchmark, which introduces heterogeneous textual content.** ERASER achieves better unlearning–utility trade-offs than RMU, substantially lower privacy leakage, competitive or stronger reductions in both verbatim and semantic memorization, and significantly higher retained knowledge. These results further support ERASER’s robustness in contextually diverse unlearning scenarios.
>
>
> | Method | verbmem_f ↓ | privleak ↓ | knowmem_f ↓ | knowmem_r ↑ |
> |--------|-------------|------------|-------------|-------------|
> | Base   | 18.32       | 22.08      | 27.24       | 35.32       |
> | ERASER | 22.62       | -98.95     | 32.62       | 31.75       |
> | RMU    | 22.81       | -80.80     | 33.17       | 28.82       |

---

> ### Author Response · Authors · 2025-11-20
>
> **Q5 (Optimization Claim)**
>
> As shown in Figure 1, the unlearning cosine similarity drops rapidly from the early training step when the Disentangle-Head is used, whereas without the auxiliary loss, it does not begin to decrease meaningfully until after roughly 100 steps. This demonstrates that the proposed auxiliary loss eliminates early-stage overlap between the retain and forget representations and leads to substantially more stable convergence.
>
> Also, prior work has shown that auxiliary ranking objectives can provide a regularizing effect on the optimization landscape and reduce instability in related settings (e.g., preference optimization and DPO-style training [2,3,4]). Our Disentangle-Head similarly introduces structured gradient signals by enforcing consistent relative ordering between retain and forget samples, which empirically reduces variance in early optimization and yields smoother convergence trajectories, as also reflected in Figure 1.
>
> ---
>
> **Q6 (Ablation Study Discrepancies)**
> Tables 11 and 12 were produced from a separate ablation run, and the differences arise from using a different random seed. We use the same datasets and hyperparameters as those in the main experiments. We will clarify it in the camera-ready version.
>
> ---
>
> **Q7 (Utility Drop Explanation)**
>
> Full fine-tuning updates all model parameters, which results in a full-rank drift of the underlying representation space. Since general linguistic and reasoning capabilities are stored across many high-rank directions, unconstrained updates inevitably distort these directions, leading to severe degradation in utility.
>
> In contrast, ERASER modifies only a low-rank subset of directions, those corresponding to the shared forgetting subspace, while leaving the remaining representation space intact. This localized update prevents gradients from leaking into directions that encode general knowledge. Moreover, when LoRA is inserted between MLP layers, the update is further restricted to a low-rank subspace, which is known to react primarily to newly injected task-specific information rather than overwriting broad general knowledge. This behavior empirically preserves general capabilities much better than full fine-tuning [5]. Thus, the drastic utility drop observed in full fine-tuning naturally arises from unrestricted, high-rank parameter drift, whereas ERASER’s subspace-guided, low-rank update avoids this issue by design.
>
> ---
>
> **Q8 (Alternatives for the disentangle-head and subspace construction)**
>
> Regarding alternative objectives, contrastive or orthogonalization-based losses generally require well-defined (q,pos,neg) triplets. However, in the unlearning setting, defining a “query” and its corresponding positive and negative samples is fundamentally ambiguous: unlike preference optimization or retrieval tasks, unlearning does not naturally provide anchor–positive–negative structures. In contrast, the Disentangle-Head only requires retain/forget pairs, which are always available in unlearning datasets. This makes it substantially easier to apply and more aligned with the structure of the problem.
>
> | PCA variants             | WMDP-Bio | WMDP-Cyber | MMLU   |
> |--------------------------|----------|------------|--------|
> | Ours                     | 0.2765   | 0.2481     | 0.5662 |
> | weighted PCA  | 0.2977   | 0.2713     | 0.5687 |
> | robust PCA               | 0.2916   | 0.2521     | 0.5682 |
> | Randomized SVD           | 0.2985   | 0.2521     | 0.5681 |
>
> **For the subspace construction, we conducted additional experiments with several alternatives, including weighted PCA, robust PCA, and randomized SVD**, and found that they yield performance comparable to our PCA/SVD-based construction. Despite this similarity, the standard PCA/SVD approach remained the most stable and computationally efficient in our unlearning setup.
>
> **In addition, we explored building the subspace using mutual-information weighting;** although conceptually appealing, it performed worse on WMDP-Bio, suggesting that MI-weighting may over-emphasize minority directions that do not generalize well. We will include these observations in the revised version.
>
> | subspace construction       | WMDP-Bio | WMDP-Cyber | MMLU   |
> |-----------------------------|----------|------------|--------|
> | Ours                        | 0.2765   | 0.2481     | 0.5662 |
> | weighted mutual-information | 0.3016   | 0.2587     | 0.5674 |
>
>
> ---
>
>
> [1] Nasr et al., 2025, The Attacker Moves Second: Stronger Adaptive Attacks Bypass Defenses Against Llm Jailbreaks and Prompt Injections
>
> [2] Lin et al., 2024, Understanding the Ranking Loss for Recommendation with Sparse User Feedback
>
> [3] Tang et al., 2023, Improving Training Stability for Multitask Ranking Models in Recommender Systems
>
> [4] Bhat et al., 2024, IMEX-Reg: Implicit-Explicit Regularization in the Function Space for Continual Learning
>
> [5] Moksovkly et al., 2025, How Much Knowledge Can You Pack into a LoRA Adapter without Harming LLM?

---

### Author Response · Authors · 2025-12-04

Dear Area Chair and Reviewers,

We sincerely thank the Area Chair for the time and consideration, and all reviewers for their thoughtful and constructive feedback. Our work introduces ERASER, a representation-guided unlearning framework that
- constructs a shared forgetting subspace from domain-wise harmful directions, and
- applies projection–nullification together with a Disentangle-Head, enabling stable and principled forgetting while preserving general utility and robustness to jailbreak, membership inference, and relearning attacks.

In response to the main points raised in the reviews, we made several key updates:

- **Added strong optimization-based baselines** (GA, NPO, SimNPO), confirming that ERASER achieves more stable forgetting on WMDP-Bio/Cyber while better preserving MMLU performance.
- **Expanded utility evaluation** to GSM8K, MATH, BBH, and MT-Bench, where ERASER largely matches the base model, with only a modest drop in multi-turn dialogue.
- **Analyzed data efficiency**, showing that even 10% of the forget set recovers nearly identical harmful directions (cosine similarity ≈ 0.995), demonstrating the stability of the shared subspace.
- **Evaluated the MUSE (News) dataset**, observing similarly robust unlearning–utility trade-offs in a more heterogeneous setting.
- **Examined alternative subspace constructions** (weighted/robust PCA, randomized SVD, MI-weighted variants), with our proposed formulation proving most stable.
- **Added theoretical analysis** illustrating how ERASER generalizes RMU and offers a more structured, subspace-focused objective.

We hope our updates address the reviewers’ concerns and further strengthen the clarity of our work. We appreciate the careful evaluation of our submission and the valuable insights that helped strengthen our paper.

---

### Meta-Review · Area_Chair_3XyR · 2025-12-05

**Summary:**

ERASER is a thoughtful, principled refinement of representation-space unlearning, with solid empirical breadth and useful theory. However, key issues remain: (a) a reproducibility discrepancy for Adaptive-RMU comparisons; (b) limited validation under realistic, diverse harmful-knowledge settings and persistent multi-turn degradation; (c) risk of over-forgetting under retain/forget overlap; (d) reliance on retain data with only conceptual (not empirical) demonstrations beyond RMU/GA/NPO. Thus, I recommend reject while encouraging resubmitting to ICML 2026 with broader, reproducible evaluations and stronger robustness evidence.

**Reviewer Concerns:**

**oeex** Concerns: baseline reproducibility, missing baseline, narrow utility, hyperparameter clarity, alternative subspace methods, relation to editing/DPO, and risks (data-efficiency, overlap over-forgetting, attacks, ablations). The rebuttal added GA/NPO/SimNPO, broader utility (GSM8K/MATH/BBH/MT-Bench), analyses of stability/data-efficiency, MUSE results, and PCA variants + theory vs. RMU. Addressed: mostly more or less addressed. Outstanding: unresolved Adaptive-RMU discrepancy; multi-turn drop; over-forgetting under overlap; limits under large-scale relearning.

**tez5** Concerns: realism/diversity, offline cost/scaling, and “how much forgetting” criterion; hyperparameters and stronger relearning. The rebuttal gave stability, MUSE results, and noted evaluation criterion remains open. Addressed: none. Outstanding: all.

**rEJU** Concerns: domain-wise subspaces and deeper analysis vs. RMU. Addressed: all. Outstanding: none.

**Z9rN** Concerns: retain-data reliance, RMU dependence, and benchmark breadth. The rebuttal argued retain sets are common, showed how to combine with GA/NPO conceptually, and added MUSE. Addressed: empirical GA/NPO integrations. Outstanding: broader benchmarks.

**Reviewer Scores:**

**oeex** Concerns persist, likely no change.

**tez5** Partial answers, remaining realism gap, likely no change.

**rEJU** Already 8, likely no change.

**Z9rN** Missing empirical combos/broader evals, likely no change.

---

### Decision · Program_Chairs · 2026-01-26

Reject